Crustacean amphipods from marsh ponds: a nutritious feed resource with potential for application in Integrated Multi-Trophic Aquaculture

Jiménez-Prada Pablo pjimenez9@us.es 1 2
Hachero-Cruzado Ismael 2
Giráldez Inmaculada 3
Fernández-Diaz Catalina 2
Vilas César 3
Cañavate José Pedro 2
Guerra-García José Manuel 4
1 Laboratorio de Biologia Marina, Departamento de Zoología, Universidad de Sevilla , Sevilla , Spain
2 Instituto de investigacion y Formacion Agraria y Pesquera, El Toruño , Puerto de Santa María , Spain
3 Departament de Química “Prof. J.C. Víchez Martín”, Facultad de Ciencias Experimentales, Universidad de Huelva , Huelva , Spain
4 Laboratorio de Biologia Marina, Departamento de Zoología, Universidad de Sevilla , Sevilla , Spain
Kaushik Sadasivam
Electronic publication date: 2018 Jan 12
Publication date: 2018
Volume: 6
Electronic Location ID: e4194
Received 2017 Aug 30; Accepted 2017 Dec 5
Copyright: ©2018 Jiménez-Prada et al.
Copyright year: 2018
Copyright holder: Jiménez-Prada et al.
License: This is an open access article distributed under the terms of the Creative Commons Attribution License, which permits unrestricted use, distribution, reproduction and adaptation in any medium and for any purpose provided that it is properly attributed. For attribution, the original author(s), title, publication source (PeerJ) and either DOI or URL of the article must be cited.
License URL: https://creativecommons.org/licenses/by/4.0/

Keywords: Alternative prey, Amino acid, Amphipods, Aquaculture, Fatty acid, Ponds, Nutrition, Marsh, Lipid classes, Trace metals

Funding: conseRjeria de Innovación, Ciencia y Empresa, Junta de Andalucía P11-RNM-7041 INIA Financial support of this work was provided by the “conseRjeria de Innovación, Ciencia y Empresa, Junta de Andalucía (project P11-RNM-7041)”, which includes a PhD grant to Pablo Jiménez-Prada. Ismael Hachero-Cruzado was supported by an INIA postdoctoral contract. The funders had no role in study design, data collection and analysis, decision to publish, or preparation of the manuscript.

==============================
Coastal protection, nutrient cycling, erosion control, water purification, and carbon sequestration are ecosystem services provided by salt marshes. Additionally, salt ponds offer coastal breeding and a nursery habitat for fishes and they provide abundant invertebrates, such as amphipods, which are potentially useful as a resource in aquaculture. Fishmeal and fish oil are necessary food resources to support aquaculture of carnivorous species due to their omega-3 long-chain polyunsaturated fatty acids (n-3 LC-PUFA). Currently, aquaculture depends on limited fisheries and feed with elevated n-3 LC-PUFA levels, but the development of more sustainable food sources is necessary. Amphipods appear to be a potential high quality alternative feed resource for aquaculture. Hence, a nutritional study was carried out for several main amphipod species—Microdeutopus gryllotalpa, Monocorophium acherusicum, Gammarus insensibilis, Melita palmata and Cymadusa filosa—in terrestrial ponds in the South of Spain. These species showed high protein content (up to 40%), high n-3 PUFA and phospholipid levels, and high levels of phophatidylcholine (PC), phosphatidylethanolamine (PE) and triacylglycerols (TAG), the latter being significantly high for M. acherusicum. M. gryllotalpa and M. acherusicum showed the highest proportion of lipids (19.15% and 18.35%, respectively). Isoleucine, glycine and alanine were the dominant amino acids in all species. In addition, amphipods collected from ponds showed low levels of heavy metals. Furthermore, the biochemical profiles of the five species of amphipods have been compared with other studied alternative prey. Therefore, pond amphipods are good candidates to be used as feed, and are proposed as a new sustainable economic resource to be used in aquaculture. G. insensibilis may be the best for intensive culture as an alternative feed resource because it shows: (1) adequate n-3 PUFA and PL composition; (2) high levels of glycine, alanine, tyrosine, isoleucine and lysine; (3) high natural densities; (4) large body size (≥1 cm), and (5) high concentration of calcium. Moreover, a combined culture of amphipods and fishes in these marsh ponds seems a promising and environmentally sustainable way to develop Integrate Multi-Trophic Aquaculture (IMTA) in these ecosystems.

Introduction

Saltmarshes are aquatic systems with many ecosystem services arising from ecosystem processes and functions such as: coastal protection, nutrient cycling, erosion control, water purification, and carbon sequestration (Barbier et al., 2011). Nowadays, approximately 50% of salt marshes suffer deterioration due to human activity (Barbier et al., 2011). Interestingly, some salt marshes, such as those located in Southern Spain, require anthropogenic activities (hydraulic control, wall conservation, sediment removal, etc.) to be economically sustainable in the context of aquaculture (Arias & Drake, 1999). These modified saltmarshes provide, therefore, services as a coastal breeding and nursery habitat, and are used by locals for the practice of a kind of extensive aquaculture generating economic activity.

The saltmarsh zone located East and South of the Parque Natural Bahía de Cádiz (PNBC), in the South of Spain, is characterized by a complex system of tidal channels and creeks that supply seawater to saltmarsh fish-ponds situated along their course. Most of the saltmarsh ponds remain permanently flooded during the greater part of the year, and constitute a semi-natural lagoon ecosystem exploited for extensive and semi-intensive fish culture (Arias & Drake, 1994). These ponds are productive ecosystems thanks to their particular hydrology and morphology, allowing for optimal use of the available light and nutrients. The marshes’ elevated primary productivity is thus associated to a significant nutrient load and the flooding capacity in the adjacent lands (Cañavate et al., 2015). The macroinvertebrate saltmarsh community is subjected to intensive predation from fish and shorebirds, and is a main food source for non-intensively reared fish (Arias & Drake, 1994). Gastropods, amphipods and chironomid larvae dominate the macrofauna in terms of abundance (Arias & Drake, 1994).

Fish accounted for 16.7% of the global population’s intake of animal protein and 6.5% of all protein consumed in 2010 (FAO, 2014; Tocher, 2015). Moreover, what is arguably of greatest importance to consumers in the developed world, fish and seafood are unique and rich sources of omega-3 (n-3) long-chain polyunsaturated fatty acids (LC-PUFA), particularly eicosapentaenoic (EPA; 20:5n-3) and docosahexaenoic (DHA; 22:6n-3) acids. An increasing proportion of fish are farmed, accounting for almost half of all fish for human food in 2012 (Tocher, 2015). Aquaculture of carnivorous species depends on n-3 LC-PUFA from fishmeal and fish oil but these come from limited, overexploited fisheries or from fish slaughter waste. Therefore, the continued growth of aquaculture will depend on the development of more sustainable feeds with alternative ingredients, generally derived from terrestrial agriculture, with important consequences for the supply of n-3 LC-PUFA (Tocher, 2015) or, as an alternative, finding new marine resources.

In that sense, amphipods could serve as an adequate alternative food resource for aquaculture. They are important low trophic position organisms which play a major role in the biological processing of algae inputs and facilitate the transfer of nutrients from the ocean to the coastline (Hamed et al., 2014). They are also regarded as important food for economically relevant fish species (Jimenez-Prada, Hachero-Cruzado & Guerra-García, 2015). Additionally, amphipods are abundant throughout the year in marsh zones, reach high densities (ca. 60.000 ind/m2), and show adequate protein content and fatty acid profiles, with high levels of beneficial polyunsaturated fatty acids (DHA and EPA) (Kolanowski, Stolyhwo & Grabowski, 2007; Hyne et al., 2009; Baeza-Rojano, Hachero-Cruzado & Guerra-García, 2014). Previous studies have explored amphipods as alternative protein sources in experimental diets for farmed fish (Moren et al., 2006; Opstad et al., 2006; Suontama et al., 2007) and cephalopods (Baeza-Rojano et al., 2012; González, Pérez-Schultheiss & López, 2011; Baeza-Rojano et al., 2013a) obtaining promising results.

With the aim to select the more adequate amphipods from terrestrial ponds as alternative prey, we studied the composition (protein, carbohydrates, lipids and ashes) of the dominant species. Furthermore, lipid classes, fatty acids, amino acids and trace/major elements were also studied in depth. This allowed us to compare with other alternative preys used in aquaculture, in addition to evaluating their nutritional composition and potential to be used as alternative feed in aquaculture.

Materials and Methods

Sample collection

Amphipods were collected at terrestrial marsh ponds located at IFAPA “El Toruño” (Fig. 1), El Puerto de Santa María, South Spain, within the “Bahía de Cádiz” Natural Park. Six ponds were selected (Fig. 1A) for setting passive traps at shore areas during 1 week in December 2014. Salinity ranged from 23.6 to 31.8 and temperature from 12.0 to 15.4 °C. Passive traps, made up of filling oyster’s cylindrical mesh structures with nylon mesh poufs, were used to collect amphipods (Fig. 1B). Traps were fixed close to or inside algae patches of Ulva spp.

Figure 1 Ponds of IFAPA “El Toruño”.

(A) Map of ponds of IFAPA “El Toruño” with samples stations. (B) Traps were used to collect amphipods. (C) Ponds.

The studied species were those found at high densities in the traps, and which are common in macroalgae beds (mainly Ulva spp) at marsh ponds: Gammarus insensibilis Stock 1966, Melita palmata Montagu 1804, Cymadusa filosa Savigny 1816 Microdeutopus gryllotalpa Costa, 1853, and Monocorophium acherusicum Costa 1853 as described by Arias & Drake (1999). Details on date and collection sites together with size and additional information for each species are given in Table 1.

Because species’ composition and density differed among ponds, in order to have enough biomass for analytical procedures, for each goal species two pools of specimens were selected from the two stations where this species was present at the highest densities. Results are shown as the mean value for the two pools at each station. Specimens were identified to the species level with a binocular stereomicroscope, cleaned with distillated water and directly frozen at −80 °C for later analysis.

Biochemical analyses

Ash content

The ash content was gravimetrically determined after burning up at 550 °C for 4 h in a muffle furnace.

Carbohydrates

To estimate the total carbohydrate content 100 microliters of homogenised samples (5 mg/ml) were included in 900 microliters, 25 microliters of phenol (81%) and 2.5 ml of sulphuric acid (95–98%), shaken with vortex and maintained in darkness for 30 min; after this time, absorbance was measured at 485 nm (Duboi et al., 1956) following Kochert’s techniques (Kochert, 1978).

Table 1 Characteristic of the five studied species.

Information of Arias & Drake (1999). Site numbers correspond with numbers of Fig. 1A.

Species	Gammarus insensibilis	Melita palmata	Cymadusa filosa	Microdeutopus gryllotalpa	Monocorophium acherusicum	
Site	1	6	5	2	4	2	4	3	1	5	
Collection date	15∕12∕2014	18∕12∕2014	22∕12∕2014	15∕12∕2014	22∕12∕2014	15∕12∕2014	22∕12∕2014	22∕12∕2014	15∕12∕2014	22∕12∕2014	
Size	>1 cm	>1 cm	>1 cm	<0.5 cm	<0.5 cm	
Other characteristic	Associated to macroalgae, high density in spring (>3,000 ind/m2)	Large size range, max. density in winter (>1,000 ind/m2)	Associated to macroalgae, max. density in winter (300 ind/m2)	High density in winter (>5,000 in/m2)	High density in winter (>14,000 ind/m2), create a fang tube to live	

Total protein and amino acid composition

Total protein analyses were determined using the Lowry method (Lowry et al., 1951) increasing the concentration of NaOH (1.5 M) and heating at 100 °C for 60 min.

Acid hydrolysis was used to release individual amino acids from peptide and protein samples: 5–10 mg of homogenized samples of biota were placed in vials with 0.5 mL of 6 M HCl. Vials were flushed with N2, sealed with PTFE-tape and a heat-resistant cap, and placed in an oven at 150 °C for 70 min. After hydrolysis, the samples were dried in a heating block at 60 °C under a gentle stream of N2, and the remaining solids were re-suspended in 200 µL 0.1 M HCl and stored at −20 °C (Walsh, He & Yarnes, 2014). An aliquot (100 µL) of standard solution or sample was placed in a 2 mL vial, adding 400 µL of a water:ethanol:pyridine (60:32:8) mixture and 40 µL of ethyl chloroformate. It was capped and vigorously shaken using a vortex mixer for 30 s at room temperature. Gas evolution (carbon dioxide) usually occurs. Then, 200 µL of chloroform (containing 1% ECF) were added and the derivatives were extracted into the organic phase by striking the tube against a pad for about 30 s. The organic phase was dried with anhydrous sodium sulphate. The organic layer was transferred into a new vial with a 300 µL fixed insert. Aliquots (1 µL) of the derivatized extracts were injected into a Shimadzu GC-MS (GCMS-TQ8030) equipped with an Agilent HP-5MS fused silica capillary column (30 m × 0.25 mm i.d., 0.25 µm film thickness). The gas chromatography system was equipped with a split/splitless injection port operating in Splitless mode. The oven temperature was programmed from 40 °C (5 min) to 270 °C (20 min), increasing the temperature at a rate of 5 °C min−1. The transfer line was heated at 280 °C. The carrier gas was helium with a constant flow of 1 mL min−1 (mean velocity 36 cm s−1). The mass spectrometer was performed with electron ionization (EI) at 70 eV, operating in scan mode (75–500 amu). Identification of derivative amino acids was achieved by comparing the gas chromatography retention times and mass spectra with those of the pure standard compounds. All mass spectra were also compared with the data system library (NIST 11).

Total lipids, lipid classes and fatty acids

Total lipids (TL) were extracted with chloroform:methanol (2:1 v/v) containing 0.01% of butylated hydroxytoluene (BHT) as an antioxidant (Christie, 2003). The organic solvent was evaporated under a stream of nitrogen and the lipid content was determined gravimetrically. Lipid classes were separated by one dimensional double development high performance thin layer chromatography (HPTLC) using methyl acetate/isopropanol/chloroform/methanol/0.25% (w/v) KCl (25:25:25:10:9 by vol.), as the polar solvent system and hexane/diethyl ether/glacial acetic acid (80:20:2 by vol.), as the neutral solvent system. Lipid classes were visualized by charring at 160 °C for 20 min after dipping in cupric acetate in 3% phosphoric acid (Olsen & Henderson, 1989). Final quantification was made by densitometry in a CAMAG scanner at a wavelength of 325 nm, and by comparison with an external standard (Sigma-Aldrich). To quantify fatty acids (FA) the TL extracts were subjected to acid-catalyzed transmethylation for 16 h at 50 °C, using 1 ml of toluene and 2 ml of 1% sulphuric acid (v/v) in methanol. FAME were separated and quantified using a Shimadzu GC 2010-Plus gas chromatograph equipped with a flame ionization detector (280 °C) and a fused silica capillary column SUPRAWAX280 (15 m × 0.1 mm I.D.). Hydrogen was used as the carrier gas and the oven initial temperature was 100 °C for 0.5 min., followed by an increase at a rate of 20 °C min−1 to a final temperature of 250 °C for 8 min. Individual FAME were quantified by comparison with external standards (Sigma-Aldrich, St. Louis, MO, USA) (Guerra-García et al., 2016).

Major and trace elements

Approximately 10 mg of the sample was digested with 0.5 mL of conc. HNO3 and 1.0 mL of 30% H2O2 . Then final solutions were made up to 5.0 mL in a volumetric flask with Milli-Q water. Trace and major elements concentrations were analyzed by ICP-MS (7700; Agilent, Santa Clara, CA, USA) and ICP-OES (Ultima2; Jobin Yvon, New York, NY, USA), respectively, at the University of Huelva. Multi-element calibration standards were freshly prepared by dilution from certified stock solutions, standard solutions of ultrapure quality, and milliQ-water. The accuracy and precision of the measurements was greater than 3% RSD.

Statistical analyses

The gross biochemical composition of the amphipod species (protein, lipid, ash, and carbohydrate) was expressed as the percentage on a dry weight basis.

To explore potential differences in the contribution of ashes, proteins, carbohydrates and total lipids, one-way ANOVA was used, including species as factor, with the two values of each station as replicates (n = 2). Prior to the ANOVAs, the homogeneity of variances was tested with Cochran’s test. Where variances remained heterogeneous, even with transformation, untransformed data was analysed, as ANOVA is a robust statistical test and is relatively unaffected by the heterogeneity of variances, particularly in balanced experiments (Underwood, 1997). In such cases, to reduce type I error, the level of significance was reduced to <0.01. When ANOVA indicated a significant difference, the source of the difference was identified using the Student–Newman–Keuls (SNK) tests.

Principal component analysis (PCA) was conducted to lipid class, fatty acids, amino acids and trace/major elements matrices for the ordination of amphipod species. Additionally, permutation tests for multivariate analysis of similarity (PERMANOVA) were conducted to explore significant differences among species using the different matrices (lipid classes, fatty acids, amino acids and trace/major elements).

Univariate analyses were conducted with GMAV, and multivariate analyses were carried out using PRIMER6 and PERMANOVA+ package (Clarke & Gorley, 2001).

Results

General composition

The five dominant species in the present study (Table 1) showed a similar composition of ash, proteins and carbohydrates, but differed slightly in total lipids (Table 2 and Fig. 2). SNK tests revealed that M. gryllotalpa and M. acherusicum had a higher proportion of lipids (19.15% ± 0.48 and 18.35 ± 0.23, respectively, mean ± standard deviation) than the other three species, G. insensibilis (12.98 ± 2.01), M. palmata (15.9 ± 2.50) and C. filosa (13.42 ± 1.68). In general terms, all the species were characterised by high levels of protein (30.85–39.58%) and ash (34.81–49.34%) and lower levels of carbohydrates (3.51–6.90%) and lipids (11.56–19.48%).

Table 2 Results of ANOVAs for general composition (Ash, Proteins, Carbohydrates and Total lipids).

	Source of variation	df	MS	F	P	F versus	
Ash	Species	4	9.1044	0.40	0.8023	Res	
Residual	5	22.765				
Total	9					
Cochran’s C-test	C = 0.9276 (p < 0.05)	
Transformation	None	
Proteins	Species	4	17.5789	4.27	0.0716	Res	
Residual	5	4.1175				
Total	9					
Cochran’s C-test	C = 0, 8409 (Not significant)	
Transformation	None	
Carbohydrates	Species	4	4.2886	1.52	0.3254	Res	
Residual	5	2.8289				
Total	9					
Cochran’s C-test	C = 0.9238 (p < 0.05)	
Transformation	None	
Total Lipids	Species	4	15.5938	5.82	0.0402*	Res	
Residual	5	2.6809				
Total	9					
	C = 0.4663 (Not significant)	
Transformation	None	
Notes.

* indicates significant differences at p < 0.05.

Figure 2 Percentage of dry weight of proteins, carbohydrates, lipids and ash.

Dry weight percentage of proteins, carbohydrates, lipids and ash per species and station.

Figure 3 PCA of amphipods’s amino acids.

Principal Component Analysis (PCA) plot based on the Amino acid composition of the five amphipod species. GI, Gammarus insensibilis; MP, Melita palmata; CF, Cymadusa filosa; MG, Microdeuteropus gryllotalpa; MA, Monocorophium acherusicum. The numbers after the species indicate the collection station. Only variables significantly correlated with axis 1 and 2 are represented.

Amino acids

The 13 amino acids detected with acid extraction are shown in Table 3. The seven most abundant amino acids accounted for about 75% of total amino acids. Among them, three are essential (isoleucine, lysine and valine), three are non-essential (glycine, alanine and aspartic acid) and one is a semi essential amino acid (tyrosine).

PERMANOVA analysis did not show differences in amino acids among species (Pseudo-F = 2.4224, P = 0.061). Axis 1 of PCA analysis (Fig. 3) explained 57.1% of total variation; tyrosine (r = 0.615, p < 0.05) and aspartic acid (r =  − 0.502, p < 0.05) correlated with axis 1, and separated G. insensibilis from the remaining species by higher values of tyrosine and lower aspartic acid content. The axis 2 (29.8% of total variation) correlated positively with glutamic acid (r = 0.612, p < 0.05), and separate M. acherisicum and G. insensibilis to M. palmata and C. filosa due to lower values of glutamic acid. Four amino acids showed differences (tyrosine (F = 29.905, p = 0.001), lysine (F = 14.415, p = 0.006), leucine (F = 10.774, p = 0.013) and proline (F = 29.271, p = 0.001)) among species in ANOVAs tests. SNK analysis separated G. insensibilis from the other four species with higher tyrosine and lysine, but lower proline contents; however, regarding leucine content, only M. acherusicum is separated due to its lower level.

Lipid classes and fatty acids

Regarding lipid classes, all studied species showed high PC, PE and TAG levels (Table 4). Although PERMANOVA analysis showed differences in the lipid class profile among species (Pseudo-F = 4.01, p = 0.046), pair-wise tests did not show any differences. Axis 1 of the PCA analysis (Fig. 4) explained 95.2% of total variance. TAG correlated significantly with this axis (r =  − 0.98, p < 0.01) and separated M. acherusicum, with higher values of TAG, from the other species. Axis 2 explained 3.3% of total variance. PC correlated positively with this axis (r = 0.846, p < 0.01) and separated G. insensibilis, with lower values of PC, from the other species. ANOVAs confirmed differences among species in PC (F = 18.7, p = 0.033) and TAG (F = 5.47, p = 0.045). In both lipids class, the SNK test did not show differences among species.

Figure 4 Principal Component Analysis (PCA) plot based on the Lipid Classes (LCs) composition of the five amphipods species.

PC, Phophatidylcholine; PE, Phosphatidylethanolamine; PI, Phosphatidylinositol; PS, Phosphatidylserine; TAG, Triacylglycerols; Cho, Cholesterol; GI, Gammarus insensibilis; MP, Melita palmata; CF, Cymadusa filosa; MG, Microdeuteropus gryllotalpa; MA, Monocorophium acherusicum. The numbers after the species indicate the collection station. Only variables significantly correlated with axis 1 and 2 are represented.

Table 3 Essential (E) and Not Essential (NE) amino acids.

Essential (E) and Not Essential (NE) amino acids (% of identified amino acids) of the five studied species. The values are a mean of two replicates. Each replicate consists of a pool of specimens. Numbers indicate the collection sites.

Amino acid	E/NE	G. insensibilis	M. palmata	C. filosa	M. grillotalpa	M. acherusicum	
		1	6	5	2	4	2	4	3	1	5	
Isoleucine	E	10.34	10.56	10.45	10.26	11.05	–	10.28	11.62	11.82	11.18	
Leucine	E	7.86	6.96	7.98	7.32	6.87	–	7.09	6.77	5.56	5.47	
Lysine	E	10.62	10.47	9.91	9.51	9.53	–	9.54	9.76	9.52	9.31	
Methionine	E	2.73	2.10	2.36	2.21	1.45	–	2.14	2.68	2.35	2.26	
Phenylalanine	E	6.63	6.24	6.40	6.22	6.35	–	6.35	6.48	6.24	6.42	
Valine	E	8.41	8.15	8.54	8.21	8.42	–	8.04	8.54	8.33	7.81	
Alanine	NE	10.89	11.19	10.74	10.79	11.41	–	10.77	12.05	12.43	12.46	
Aspartic acid	NE	6.79	9.52	9.66	9.98	10.54	–	10.29	9.16	10.22	10.12	
Glutamic acid	NE	2.43	2.43	3.20	4.20	3.78	–	4.64	1.50	3.01	3.34	
Glycine	NE	13.38	14.05	12.75	13.39	13.90	–	12.79	14.38	13.98	14.79	
Proline	NE	1.54	1.65	1.71	1.95	2.25	–	2.11	2.18	2.40	2.41	
Serine	NE	5.43	4.68	5.40	5.38	5.15	–	5.88	4.95	4.49	5.07	
Tyrosine	NE	12.95	12.00	10.90	10.57	9.31	–	10.07	9.93	9.64	9.37	
Total E		46.59	44.48	45.63	43.73	43.67		43.43	45.85	43.82	42.44	
Total NE		53.41	55.52	54.37	56.27	56.33		56.57	54.15	56.18	57.56	

In connection with fatty acid composition, the most abundant fatty acids in all species were: the saturated 16:0, the monounsaturated 16:1n9 and 18:1n9, and the polyunsaturated 18:2n6, 20:5n3 (EPA) and 22:6n3 (DHA) (Table 5). PERMANOVA also reflected global differences in fatty acid composition among species (pseudo-F = 9.97, p = 0.001) despite no differences in species pair-wise comparisons being detected. These differences were also supported by the PCA analysis (Fig. 5). Axis 1 explained 67.7% of total variance and axis 2 explained 22.7%. The fatty acid 22:6 n3 (DHA) correlated positively with axis 1 (r = 0.657, p < 0.05), and separated M. acherusicum and M. gryllotalpa from the other species. 20:5n3 (EPA) correlated positively with axis 2 (r = 0.59, p < 0.05), and separated M.gryllotalpa from M. acherisicum. ANOVAs confirmed significant differences for these fatty acids among species (EPA, F = 6.03 p = 0.0375; DHA, F = 53.26 p = 0.0003). For both fatty acids the SNK test did not show differences among species.

Table 4 Lipid classes (µg/100 µg of DW) per station of the five studied species per station.

The values are a mean of two replicates. Each replicate consists of a pool of specimens. Numbers indicate the collection sites.

	G. insensibilis	M. palmata	C. filosa	M. grillotalpa	M. acherusicum	
	1	6	5	2	4	2	4	3	1	5	
PC	1.96	1.75	2.36	2.52	2.19	1.93	2.36	2.54	3.08	2.90	
PE	1.27	1.17	1.58	1.66	1.52	1.29	1.52	1.49	1.72	1.50	
PI	0.17	0.18	0.31	0.31	0.33	0.23	0.15	0.11	0.13	0.32	
PS	0.26	0.36	0.40	0.46	–	0.31	0.43	–	–	0.36	
TAG	3.01	2.93	3.87	2.44	2.76	1.86	1.97	3.97	5.58	7.19	
Cho	0.32	0.18	0.35	0.25	0.30	0.35	0.24	0.25	0.332	0.30	
Notes.

PC Phophatidylcholine

PE Phosphatidylethanolamine

PI Phosphatidylinositol

PS Phosphatidylserine

TAG Triacylglycerols

Cho Cholesterolx

Table 5 Fatty acids (µg/mg of DW) of the five species studied per station.

The values are a mean of two replicates. Each replicate consists of a pool of specimens. Numbers indicate the collection sites.

	G. insensibilis	M. palmata	C. filosa	M. grillotalpa	M. acherusicum	
	1	6	5	2	4	2	4	3	1	5	
Saturated (Sat)	
14:0	1.21	1.18	1.17	1.08	1.15	1.32	1.36	1.75	1.61	1.65	
16:0	5.86	6.00	5.14	5.55	3.88	4.26	5.99	8.55	8.86	8.97	
17:0	–	–	0.73	0.50	–	–	0.64	0.68	0.64	0.62	
18:0	1.71	1.82	2.62	2.44	1.60	1.50	2.39	2.59	2.45	2.43	
Total	8.79	9.01	9.66	9.56	6.63	7.08	10.37	13.56	13.55	13.66	
Monounsaturated (MUFA)	
16:1n9	2.40	3.43	1.28	1.61	0.76	0.86	2.27	4.47	1.67	1.66	
17:1	0.45	0.60	0.72	0.47	–	–	0.58	0.65	0.64	0.62	
18:1n9	4.94	5.58	5.42	4.36	4.19	5.62	3.79	4.35	5.22	6.01	
20:1n9	0.43	0.45	0.56	0.50	0.67	0.80	0.52	0.85	0.60	0.61	
Total	8.22	10.05	7.99	6.95	5.61	7.28	7.16	10.32	8.13	8.91	
Poliunsaturated (PUFA)	
18:2n6	4.75	2.16	1.73	2.23	2.71	2.61	1.72	1.65	2.27	2.23	
18:3n6	0.44	0.48	–	–	–	–	0.66	0.85	–	–	
18:3n3	0.57	1.17	0.81	0.65	1.25	2.01	0.69	0.63	1.82	1.74	
20:2n6	0.77	0.49	0.52	0.54	0.90	0.78	–	–	0.63	0.65	
20:4n6 ARA	1.40	1.47	2.08	2.28	1.71	1.47	1.99	1.94	1.41	1.43	
20:3n3 DPA	0.36	0.41	0.36	0.40	0.62	0.92	0.39	0.41	1.02	1.03	
20:5n3 EPA	5.02	6.03	5.41	7.03	3.47	3.63	7.26	9.98	6.47	6.77	
22:6n3 DHA	3.25	3.59	5.68	4.41	2.04	1.97	4.04	4.86	8.71	8.79	
Total	16.57	15.80	16.59	17.54	12.70	13.38	16.76	20.31	22.33	22.63	
Notes.

ARA Arachidonic acid

EPA Eicosapentanoic acid

DPA Docosapentanoic acid

DHA Docosaheanoic acid

Figure 5 Principal Component Analysis (PCA) plot based on the Fatty acids composition of the five amphipods species.

20:5n3 (EPA), 22:6n3 (DHA). GI, Gammarus insensibilis; MP, Melita palmata; CF, Cymadusa filosa; MG, Microdeuteropus gryllotalpa; MA, Monocorophium acherusicum. The numbers after the species indicate the collection station. Only variables significantly correlated with axis 1 and 2 are represented.

Trace and major elements

Silver (Ag), arsenic (As), cadmium (Cd), cobalt (Co), nickel (Ni), lead (Pb) and selenium (Se) were not detected. Cr values ranged from 8.74 to 81.62 ppm, Cu from 36.42 to 162.63 ppm and Zn from 36.64 to 322.61 (Table 6). PERMANOVA results showed differences in the composition of trace and major elements among species (pseudo-F = 9.36, p = 0.007). In the PCA analysis (Fig. 6), calcium (Ca) was significantly correlated (r =  − 0.98, p < 0.001) with axis 1, which explained 96.7% of the variance; this axis separated M. acherusicum and M. palmata with lower Ca levels. Axis 2 explained 1.9% of the variance; Na (r =  − 0.60, p < 0.05) and P (r =  − 0.56, p < 0.05) were negatively correlated. ANOVA also indicated significant differences for Ca (F = 20.31, p = 0.0027). The SNK test separated two groups, where M. acherusicum and M. palmata were the group with a lower Ca level. Na (F = 1.68, p = 0.28) and P (F = 0.19, p = 0.93) did not present differences according to the ANOVA test.

Table 6 Trace and major elements (µg/g of DW) of the five studied species per station.

The values are a mean of two replicates. Each replicate consists of a pool of specimen. Detection limit of trace elements = 33.33 µg/g of DW. Detection limit of Major elements = 83.33 µg/g of DW.

	G. insensibilis	M. palmata	C. filose	M. grillotalpa	M. acherusicum	
	1	6	5	2	4	2	4	3	1	5	
Trace elements	
Ag	nd	nd	nd	nd	nd	nd	nd	nd	nd	nd	
As	nd	nd	nd	nd	nd	nd	nd	nd	nd	nd	
Cd	nd	nd	nd	nd	nd	nd	nd	nd	nd	nd	
Co	nd	nd	nd	nd	nd	nd	nd	nd	nd	nd	
Cr	16.23	12.30	8.74	15.10	8.98	9.27	27.11	81.62	14.28	14.08	
Cu	87.04	133.73	38.76	126.92	134.90	52.88	78.02	162.63	52.11	36.42	
Ni	nd	nd	nd	nd	nd	nd	nd	nd	nd	nd	
Pb	nd	nd	nd	nd	nd	nd	nd	nd	nd	nd	
Se	nd	nd	nd	nd	nd	nd	nd	nd	nd	nd	
Zn	76.98	104.64	36.64	171.79	53.31	44.96	57.38	322.61	111.37	45.69	
Major elements	
Al	6,870	7,209	11,880	8,103	6,364	6,755	6,792	7,019	9,611	12,236	
Ca	247,771	248,136	195,680	192,075	223,772	257,195	256,192	275,224	158,634	172,823	
K	17,197	17,327	9,136	16,778	10,173	17,700	15,643	14,008	9,809	8,474	
Fe	1,805	949	1,909	1,289	837	1,316	2,114	2,220	1,566	1,814	
Mg	13,123	14,098	15,078	15,811	17,111	13,852	18,693	20,532	13,396	13,944	
Na	31,793	36,698	20,066	27,681	26,060	36,693	30,951	37,165	28,885	21,476	
P	14,240	21,992	17,343	19,671	14,073	26,080	17,002	16,278	21,949	18,223	
S	18,251	18,691	17,992	20,644	24,265	20,945	29,232	26,668	12,616	12,263	
Notes.

nd under detection limit

Figure 6 Principal Component Analysis (PCA) plot based on the Trace/Major elements composition of the five amphipods species.

GI, Gammarus insensibilis; MP, Melita palmata; CF, Cymadusa filosa; MG, Microdeuteropus gryllotalpa; MA, Monocorophium acherusicum. The numbers after the species indicate the collection station. Only variables significantly correlated with axis 1 and 2 are represented.

Discussion

Aquaculture development requires sources of marine protein, n-3 LC-PUFA and micronutrients that are more sustainable than those currently in use, because, according to the FAO’s (2016) report, fishery activity has stabilized around 90 million tons in the last 20 years, while aquaculture activity reached 106 million tons in 2015, exceeding fishery activity by 12.3 million tons. Because of that, it is necessary to investigate new marine feed sources to be used in aquaculture. Previous works have explored the use of alternative preys, like copepods, and traditional preys, like Artemia and rotifers (Tables 7–9). Amphipods from marsh ponds could also be an alternative source. To date, amphipods have been used in meals for finfish with different results. Themisto libellula (Amphipoda: Hyperiidae) has been used in different studies: (1) Moren et al. (2006) found improved growth (salmon) with a total substitution of fish meal for amphipod meal; (2) Suontama et al. (2007) described that a substitution lower than 60% did not affect post-mortem pH and rigidity in trout; however, (3) Opstad et al. (2006) obtained negative results in meals with higher than 25% substitution for feeding Atlantic cod. Another amphipod, Jassa marmorata, was used as live prey in a cephalopod species (Robsonella fontaniana) achieving 20% higher growth than with Artemia (González, Pérez-Schultheiss & López, 2011). In Baeza-Rojano et al. (2012), Sepia officinalis had a similar growth using amphipods or mysids, and, in Baeza-Rojano et al. (2013a), marine gammarideans were better live prey than Artemia and freshwater gammarids for culture of Octopus maya. Moreover, Lari et al. (2013) showed that sturgeon species, Acipenser gueldensuedtii and A. ruthenus, preferred Gammarus to polychaetes of the genus Nereis.

Table 7 Amino acid comparison of amphipods with others prey.

Essential (E) and Non-Essential (NE) amino acids (% of identified amino acids) of other prey.

Taxa	Species	Amino acids	References	
		Essencials	Non essencials		
		Isoleucine	Leucine	Lysine	Methionine	Phenylalanine	Valine	Alanine	Aspartic acid	Glutamic acid	Glycine	Proline	Serine	Tyrosine		
Amphipod	Gammarus insensibilis	10.45	7.41	10.55	2.41	6.44	8.28	11.04	8.16	2.43	13.72	1.60	5.06	12.47	Present study	
Amphipod	Melita palmata	10.35	7.65	9.71	2.28	6.31	8.37	10.76	9.82	3.70	13.07	1.83	5.39	10.74	Present study	
Amphipod	Cymadusa filosa	11.05	6.87	9.53	1.45	6.35	8.42	11.41	10.54	3.78	13.90	2.25	5.15	9.31	Present study	
Amphipod	M. gryllotalpa	10.95	6.93	9.65	2.41	6.41	8.29	11.41	9.73	3.07	13.59	2.15	5.42	10.00	Present study	
Amphipod	M. acherisicum	11.50	5.51	9.41	2.30	6.33	8.07	12.45	10.17	3.18	14.39	2.40	4.78	9.50	Present study	
Copepod	Pseudodiaptomas inopinus	4.89	9.28	8.90	2.53	5.41	6.31	8.44	11.40	17.31	7.68	6.60	5.58	5.67	Yang & Hur (2014)	
Copepod	Paracyclopina nana	5.26	9.69	8.94	0.62	5.98	6.48	7.98	12.17	17.82	6.40	6.79	5.67	6.22	Yang & Hur (2014)	
Branchiop	Artemia spp.	6.09	10.07	10.97	1.97	5.70	6.30	7.24	10.76	16.07	6.26	7.97	6.99	3.60	Yang & Hur (2014)	
Rotifers	Brachionus plicatilis	5.56	10.37	8.36	0.42	6.77	5.20	5.84	10.96	16.24	4.61	14.59	7.41	3.66	Yang & Hur (2014)	
Copepod	Pool of copepods	4.40	7.60	7.40	2.30	4.10	5.30	7.10	9.60	13.60	7.50	5.30	5.30	4.60	Hamred et al. (2013)	

Table 8 Lipid classes comparison of amphipods with others prey.

Comparison of lipid classes (µg/100 µg of DW). Taxa are organized according to increasing levels of Total Phospholipids (Total PL).

Taxa	Species	TAG	PC	PE	PI	PS	Total PL	References	
Amphipod	G. insensibilis	2.97	1.86	1.22	0.17	0.31	3.56	Present study	
Amphipod	C. filosa	2.31	2.06	1.41	0.28	0.15	3.90	Present study	
Branchiopod	Artemia franciscana	17.84	1.74	1.43	0.42	0.32	3.91	Van der Meeren et al. (2008)	
Amphipod	M. grillotalpa	2.97	2.45	1.51	0.13	0.21	4.30	Present study	
Rotifer (ME)	Brachionus sp.	–	2.50	1.90	–	–	4.40	Li et al. (2015)	
Rotifer (emulL)	Brachionus sp.	–	2.20	2.10	–	–	4.41	Li et al. (2015)	
Zooplankton	–	4.21	1.90	1.52	0.56	0.48	4.46	Van der Meeren et al. (2008)	
Amphipod	M. palmata	3.16	2.44	1.62	0.31	0.43	4.80	Present study	
Copepods	–	2.63	2.00	1.99	0.39	0.55	4.93	Van der Meeren et al. (2008)	
Amphipod	M. acherusicum	6.38	2.99	1.61	0.22	0.18	5.01	Present study	
Rotifer	Brachionus plicatilis	6.06	1.85	2.10	1.06	0.53	5.54	Van der Meeren et al. (2008)	
Copepod	Acartia tonsa	0.00	–	–	–	–	7.00	Olsen et al. (2014)	
Notes.

PC Phophatidylcholine

PE Phosphatidylethanolamine

PI Phosphatidylinositol

PS Phosphatidylserine

TAG Triacylglycerols

Table 9 Fatty acids comparison of amphipods with others prey.

Comparison of percentage in lipids (% of DW) and fatty acids (µg/mg of DW). Taxa are organized according to increasing levels of DHA.

Taxa	Species	Lipids	ARA	EPA	DHA	Refrences	
Copepod	Pool of three speciesa	3.20	–	0.63	0.80	McLeod et al. (2013)	
Rotifer	Brachionus sp.	10.00	–	0.44	1.01	McLeod et al. (2013)	
Rotifer	Brachionus plicatilis	15.40	0.29	1.09	1.91	Van der Meeren et al. (2008)	
Amphipod	C. filosa	13.42	1.59	3.55	2.00	Present study	
Zooplankton	–	14.30	0.23	2.35	2.47	Van der Meeren et al. (2008)	
Rotifers	Brachionus sp.	9.80	0.72	3.20	3.00	Li et al. (2015)	
Copepod	P. anmandeti	16.00	0.24	1.47	3.09	Rayner et al. (2015)	
Amphipod	G. insensibilis	12.98	1.44	5.53	3.42	Present study	
Copepod	–	11.10	0.09	1.93	3.80	Van der Meeren et al. (2008)	
Amphipod	M. grillotalpa	19.15	1.96	8.62	4.45	Present study	
Branchiopod	Artemia franciscana	24.90	0.80	2.29	4.98	Van der Meeren et al. (2008)	
Amphipod	M. palmata	15.90	2.18	6.22	5.04	Present study	
Amphipod	M. acherusicum	18.35	1.42	6.62	8.75	Present study	
Rotifer	Brachionus sp.	12.70	0.60	4.00	10.00	Li et al. (2015)	
Copepod	–	7.07	–	5.17	11.76	Li et al. (2015)	
Copepod	Acartia tonsa	9.40	0.24	8.80	13.37	Olsen et al. (2014)	
Notes.

a Eurythemora affinis, Calanus finmarchicus and Microsetella norvegica (Copepods).

ARA Arachidonic acid

EPA Eicosapentanoic acid

DPA Docosapentanoic acid

DHA Docosaheanoic acid

General composition (proteins, carbohydrates, lipids and ashes)

In general terms, the studied amphipods from marsh ponds were characterised by high levels of protein and ash and low levels of carbohydrates and lipids. Protein values were similar to those found in copepods and mysids by other authors (Wang et al., 2014), but lower than those reported for other amphipods from the Bay of Algeciras (Baeza-Rojano, Hachero-Cruzado & Guerra-García, 2014). However, amphipods present a higher percentage of carbohydrates than other studied groups (e.g., Wang et al., 2014). Taking into account the National Research Council’s (NRC, 2011) indications, no dietary requirement for carbohydrates has been demonstrated in fish. However, carbohydrates are important for sparing proteins and lipids for energy provision, and for the synthesis of important compounds derived from carbohydrates (NRC, 2011).

Regarding lipid content, values of 5.9 and 7.7% of DW have been measured in Gammarus pulex and Carinogammarus roeselii respectively (Geng, 1925); similar to the Strait of Gibraltar amphipods (Baeza-Rojano, Hachero-Cruzado & Guerra-García, 2014), but lower than total lipids measured in this study. However, higher lipid levels are normally found in deep-sea amphipod species (45% in populations of Monoporeia affinis (Lehtonen, 1996)) and in Arctic and Antarctic species (27% in Onisimus affinis and 53% in Orchomonella plebs; Percy, 1979; Pearse & Giese, 1966).

On the negative side, the high ash levels present in all the analysed amphipods species could be considered inadequate for feeding fish larvae (Moren et al., 2006; Opstad et al., 2006; Suontama et al., 2007).

Amino acids

Dietary supplementation with ingredients rich in specific amino acids is beneficial, due to the crucial roles in cell metabolism and physiology of these molecules. Amino acids have several functions in fishes, such as: (1) increasing the chemo-attractive property and nutritional value of aquafeeds with low fishmeal inclusion; (2) optimizing efficiency of metabolic transformation in juvenile and subadult fishes; (3) reducing aggressive behaviours and cannibalism; (4) increasing larval performance and survival; (5) mediating timing and efficiency of spawning; (6) improving fillet taste and texture; and (7) enhancing immunity and tolerance to environmental stresses (Li et al., 2009).

The three most abundant amino acids in the analysed species are the non-essential amino acids glycine, alanine and aspartic acid. Although non-essential they have a significant role in stimulating feeding response and energy mechanisms in fishes (Li et al., 2009). Glycine participates in gluconeogenesis, sulphur amino acid metabolism, one-carbon unit metabolism and fat digestion (Fang, Yang & Wu, 2002), stimulates feed intake (Shamushaki et al., 2007), and has a critical role in the osmoregulatory responses of fishes and shellfishes (Li et al., 2009). Alanine and Aspartic acid are two of the major glucogenic precursors and important energy substrates for fish, and can also stimulate the feeding response (Li et al., 2009).

Isoleucine, leucine and valine are three of the most abundant essential amino acids in the analysed amphipods (EAA). These EAAs play important structural roles and are primarily deposited in body protein, notably in skeletal muscles (NRC, 2011). Phenylalanine and tyrosine, an essential and conditionally essential amino acid respectively, are also abundant in amphipod species (Table 3). The former can be converted to the latter, and dietary levels of both could profoundly influence pigmentation development, feed intake, growth performance, immunity, and survival of fish in natural environments (Pinto et al., 2008). Consequently, fishes’ dietary requirements for phenylalanine and tyrosine increase substantially during metamorphosis (Pinto et al., 2008).

Analysed species show higher levels of the non-essential amino acids glycine, alanine and tyrosine, but lower levels of leucine, than other amphipods (Table 7). The high levels of glycine and alanine are of special interest for fish feeding, because these amino acids are feed intake stimulators (Shamushaki et al., 2007). Kasumyam & Mikhailova (2014) and Kasumyan & Marusov (2015) showed an increase in the fish’s feed intake with alanine and glycine, respectively. Regarding essential amino acids, amphipods have higher levels of isoleucine and valine, but lower of leucine (Table 7). These three AA are of interest for fish feeding because they play important structural roles and are primarily deposited in body protein, notably in skeletal muscles. In addition, Valine is involved in the synthesis of the myelin covering of the nerves (Cowey & Walton, 1989; Brosnan & Brosnan, 2006).

Lipid classes and fatty acids

Among lipid classes, triglycerides (TAG) are the predominant form of lipid reserve, and are always mobilized before phospholipids (PL) during starvation (Sargent, Henderson & Tocher, 1989; Hachero-Cruzado et al., 2014). The larvae presumably utilize the TAG to satisfy their energy demands while PL, which play an important structural role in the cell membrane, tend to be conserved (Rainuzzo, Reitan & Olsen, 1997) and increase the efficiency of the transport of dietary fatty acids and lipids from the gut to the rest of the body (Coutteau et al., 1997; Fontagné et al., 1998; Salhi et al., 1999; Tocher et al., 2008). Studied amphipods have a similar range of phospholipids (PL) but higher TAG dispersion, with M. acherisicum and M. palmata showing higher levels of PL and TAG than other preys (Table 8).

Regarding fatty acids, DHA is one of the most important fatty acids for its key role in the formation and composition of nervous tissue and the retina (Mourente & Tocher, 1992; Bell, McEvoy & Navarro, 1996; Coutteau & Mourente, 1997). Recent studies search for new feed sources with higher levels of n-3 and n-6 PUFA. Table 9 compares the amounts of essential fatty acids in some crustacean species and rotifers with the amounts in “El Toruño” ponds’ amphipods. For example, rotifers studied by Li et al. (2015) and Acartia tonsa (copepod) studied by Olsen et al. (2014) have the highest values of DHA and high EPA values, but they have lower amounts of ARA and total lipids (7.0 and 9.4%, respectively). Furthermore, A. tonsa is not easy to collect since a ship with trawls is necessary (Olsen et al., 2014). We must also take into account that the rotifers had been enriched with a lipid emulsion (Marol E, prepared by SINTEF Fisheries and Aquaculture, Norway), based on the marine oil DHASCO (Martek Biosciences, Columbia, MD, USA), which is used as a TAG source rich in DHA (Li et al., 2015). Both methods increase the economic cost. However, M. palmata and M. acherusicum are two species with high values of essential fatty acids; while Cymadusa filosa is is the species with the worst fatty acid profile of the five studied amphipod species.

Major and trace elements

Data of major and trace elements of amphipods from the “El Toruño” ponds can be compared with recent studies conducted in nearby areas with anthropogenic influence. In Algeciras Bay (Guerra-García et al., 2009) showed that there were elevated concentrations of heavy metals in the amphipods Caprella penantis and Hyale schmidti in the inner sites, higher than those obtained for amphipods of the PNBC’s ponds.

Guerra-García et al. (2010b) studied the levels of trace elements in several species of peracaridean crustaceans (amphipods, isopods and tanaids) from different sites of Southern Spain, and reported similar concentrations of Cr and Cu, higher values of As, Ni and Pb, and a considerably higher concentration of Zn than in the present study’s amphipods. In the same area, Guerra-García et al. (2010a) also studied some major elements in peracarideans and all taxa presented similar levels of K and Mg (except amphipods with lower values) and similar Ca levels (except caprellids with higher values).

Other prey, such as rotifers and copepods (Mæhre, Hamre & Elvevoll, 2013), are characterised by a similar concentration of major elements (Ca, Mg, P y Cu) as that of the amphipods studied in the present work, but lower Fe values were measured in rotifers (unenriched and enriched).

Therefore, amphipods from ponds show better compositions of major and trace elements than other amphipods from nearby marinas (Guerra-García et al., 2010b), except for values of Fe. The elevated levels of Fe should be further investigated due to the effects of iron toxicity, including reduced growth, increased mortality, diarrhoea, and histopathological damage to liver cells in fish (NRC, 2011).

Nutritional requirements of aquaculture fishes

Marine fish larvae have a reduced ability to be fed prepared diets in early stages since they have lower digestion rates (Lauff & Hoffer, 1984; Kolkovsjki et al., 1993; Kolkovsjki, Arieli & Tandler, 1997), low digestive enzyme activity and inadequate nutrition (Kolkovsjki et al., 1993; Teshima, Ishikawa & Koshio, 2000; Lazo et al., 2000) because the digestive apparatus of a larva is short and incomplete. In addition, the live food provides hormones, or their regulators, or growth factor (Lauff & Hoffer, 1984; Baragi & Lovell, 1986), and aids in digestion (Rønnestad et al., 2007). Considering this the amphipods (live or dry) could be better than a prepared diet.

Most of the research to understand nutritional requirements has focused on lipid requirements, concretely the level of phospholipids, DHA and EPA fatty acids. Marine fish larvae need an elevated protein percentage in their food to grow. Several studies of Atlantic salmon and many species of sea bass and sea breams have indicated that their larvae need around 50% of protein (NRC, 2011), and the marine fish larvae have a need for approximately 10% of total lipids (Sargent et al., 1999). Regarding lipid requirements, phospholipids (principally PC) are the most necessary, with the optimal dietary level around 3%. Additionally, 1% of LC-PUFA n3 is required with a DHA:EPA 2:1 ratio (NRC, 2011). If studied amphipods are examined in the context of fish requirements we observe they have around 50% of protein, a range from 8 to 11% of total lipids, more than 3% of phospholipids (with an elevated PC level) and between 1% to 1.4% of LC-PUFA n-3, but they present a DHA:EPA 1:2 ratio, instead of the recommended 2:1 ratio.

Towards integrated multi-trophic aquaculture systems

During the last decade, there has been an increasing interest in the potential use of amphipods for aquaculture and ornamental aquariums. Woods (2009) conducted a comprehensive review examining aspects of the known biology and ecology of caprellid amphipods and their potential suitability as a novel marine finfish feed. In fact, he pointed out that caprellids could have a beneficial role to play in integrated coastal aquaculture, as a combined bioremediator and feed resource. Baeza-Rojano (2012) and Baeza-Rojano et al. (2013b) showed the suitability of amphipods to be included in Integrated Multi-Trophic Aquaculture (IMTA) programs; feeding on by-products of other cultivated species. Guerra-García et al. (2016) demonstrated experimentally that detritus (mainly composed of uneaten feed pellets and fish faeces released by cultured fish in fish farms and sea-cage structures) can be a nutritionally adequate and cheap feed for caprellid amphipods, providing a source of both omega-3 and omega-6 fatty acids. Therefore, these authors underlined the suitability of amphipods to be used in IMTA systems associated with the extensive culture of floating farms of fishes or molluscs, or with intensive cultures in terrestrial systems.

The present study reveals an interesting example of potential IMTA systems combining the extensive culture of fishes and amphipods associated to the marsh ponds of Southern Spain. The extensive culture of fishes (mainly Sparus aurata and Dicentrarchus labrax) carried out in these modified marsh ponds with aquaculture purposes, could be developed in the context of these IMTA systems. Amphipods are naturally cultured in high densities associated to algae and/or other substrates (such as traps or other artificial devices where amphipods can attach). They feed mainly on detritus (e.g., faeces) produced by the fishes growing in the ponds. Thus, a high and sustainable production of amphipods can be obtained. These amphipods could be useful (i) as natural food for fishes cultured in the marsh ponds, and (ii) as an additional resource to be used in aquaculture (alive or liophilized, and whole or integrated in fish feed). Taking into account that amphipods exhibit fast growth, quickly reach reproductive maturity, have short interbrood periods, and are opportunistic feeders, the use of traps (such as those in Fig. 1), artificial meshes, cages, etc, could increase the available substrate for amphipods to cling on, reproduce and grow. Once these structures have been fully colonized by the amphipods (in a few weeks), they could be withdrawn and the amphipods removed using freshwater. The ponds are easily accessible from land; and algae, traps or other devices can be placed and replaced without great effort and with low costs. Marsh ponds are, consequently, promising locations to develop environmentally sustainable IMTA systems.

Conclusions

Among the studied species, G. insensibilis may be the best to be intensively cultured as an alternative feed resource because it shows: (1) adequate n-3 PUFA and PL composition; (2) high levels of glycine, alanine, tyrosine, isoleucine and lysine; (3) high natural densities; (4) large body size (≥1 cm) and (5) high concentration of Ca. M. palmata is also an option mainly due to its high n-3 PUFA and phospholipid levels. In contrast, C. filosa is the less suitable candidate because of its higher ash content, and lower protein and PUFA levels. While M. acherusicum and M. gryllotalpa present high lipid and DHA content, their size is considerably smaller than other species so a higher number of specimens are required to reach the same biomass. Ponds of salt marshes from Southern Spain could be used as a sustainable source of amphipods, which are naturally growing in the macroalgae or sediments, and can be easily collected. The combination of stable amphipod populations, naturally inhabiting the ponds, with fish culture in ponds could allow for the establishment of multitrophic and sustainable aquaculture. In addition, it is necessary to investigate the optimal conditions for culturing gammarids in ponds or indoor tanks, and their economic profitability.

Supplemental Information

Supplemental Information 1 Raw data of proteins, ashes, carbohydrates and total lipids separated by species and stations

Click here for additional data file.

Additional Information and Declarations

Competing Interests

Author Contributions

Data Availability

The authors declare there are no competing interests.

Pablo Jiménez-Prada conceived and designed the experiments, performed the experiments, analyzed the data, wrote the paper, prepared figures and/or tables.

Ismael Hachero-Cruzado conceived and designed the experiments, performed the experiments, analyzed the data, wrote the paper, reviewed drafts of the paper.

Inmaculada Giráldez performed the experiments, analyzed the data, contributed reagents/materials/analysis tools, reviewed drafts of the paper.

Catalina Fernández-Diaz performed the experiments, contributed reagents/materials/analysis tools, reviewed drafts of the paper.

César Vilas and José Pedro Cañavate contributed reagents/materials/analysis tools, reviewed drafts of the paper.

José Manuel Guerra-García conceived and designed the experiments, analyzed the data, wrote the paper, reviewed drafts of the paper.

The following information was supplied regarding data availability:

The raw data has been provided as a Supplemental File.

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
