# Peer review of "Crustacean amphipods from marsh ponds: a nutritious feed resource with potential for application in Integrated Multi-Trophic Aquaculture"

_PeerJ, doi:10.7717/peerj.4194_

## Round 0.1 · original submission · Major Revisions

The MS has been reviewed by two independent referees. Both consider the MS to be of interest but raise several points which require your attention.

A point which needs also to be taken into account is the possible accumulation of "undesirable compounds or contaminants" from such an environment.

Reviewer 1 ·

Basic reporting

- The English language should be improved throughout the manuscript to ensure a clear understanding of the information by an international audience.
Some examples where the language could be improved include lines 20, 24, 28, 35-36, 49, 51, 60-70, 74,78, 103, 105, 123, 324, 327, 332, 339, 345, 385, 433, 437,443, 682, 689, 696, 703, 708.
Phrasing employed and sentence construction in the following lines makes the comprehension difficult. 16-18, 51-53, 80-82, 336-337, 341-343,367-372, 389-391, 422-425,443-445.
- The structure of the introduction as well as the background description does not clearly demonstrate how the work fits into the broader field of knowledge. Moreover the literature referring specifically to the field of study is limited and should be further developed to include references related to various aquaculture species nutritional requirements.
- The structure of the article is conform to an acceptable format of ‘standard sections’
- Figures are relevant to the content of the article, of sufficient resolution, and appropriately described and labeled.
- The raw data provided does not appear as machine-readable format as it appears as table and figure.

Experimental design

- The manuscripts describes original primary research within Aims and Scope of the journal.
- The research hypothesis is relevant & meaningful for the development of sustainable aquaculture practices, however it is not clearly defined and does not clearly state the aims to fill a given knowledge gap.
- Ethic statement is not provided and might not be applicable considering the field of study.
- The biochemical analysis and statistical analyses methods are clearly described, however the general experimental design is questionable to reach the proposed conclusions taking into account the restricted sampling period (1 week) and the number of replicates (n=2).

Validity of the findings

- The data on which the conclusions are based is robust and statistically sound within the restricted limits of the study, however it can be considered incomplete to reach the proposed conclusions given the limited time during which the samples were collected. Considering the recognized effects that environmental parameters and seasons have on the characteristics and specificity of the species studied, analysis of various replicates at different sampling periods would be required to support discussion, conclusions and recommendations about the species to be intensively cultures as alternative feed sources.
- Part of the discussion focusing on the general field of study is limited and should be further developed to illustrate diverse potentials of alternative sustainable feed sources for aquaculture production.
-Generally the discussion is referring to studies performed out of the scope of the field of study of the research undertaken making the global comprehension difficult.
- Part of the discussion referring to IMTA is not plainly justified considering that it is not clearly stated that the experimental samples were collected within IMTA systems.
- L 311: Please indicate the larval species the text is referring to.
- L 362-363: Please provide the references.

Additional comments

The manuscript is of interest considering the quest to provide sustainable alternative feed sources for the further development of aquaculture, however it requires significant major revisions and various points (indicated in the review) which should be improved upon before acceptance.

Reviewer 2 ·

Basic reporting

The article is in general well written.

Experimental design

The mean values are a result of only two measurements (n=2). However since there are no experimental treatments (this is just a decription of the composition), and each sample consists of many individuals I consider this acceptable.

Validity of the findings

The authors present the idea as a novelty, however this is not new. In pond culture (often freshwater), this is an often used technique in extensive culture systems. Whether this will be a realistic scenario for large scale production remains to be seen however that is not in the scope of this article. The data seems to be robust and is discussed in the light of previous studies.

The PCA plots are mentioned in the results, however do not come back in the discussion. I do not see the clear purpose of these plots and suggest to only include the tables.

Additional comments

Major comments:
The authors use the terms ‘trace/major metals’, ‘trace elements and metals’ trace and major elements’, etc. I would recommend to use elements for both the trace and the major ones. Many of the elements mentioned are not metals.
The authors describe that the prey collected in these ponds are lower in heavy metals than prey caught in other locations. The abundance of Ulva sp. could be a reason for this as this green algae is known to readily accumulate heavy metals and thus might lower the concentration in the water. (This is just a comment and does not have to be included in the article).


Minor comments:
Keywords
I am not sure what ‘major metal’ as a keyword means

Abstract
Line 33: change ‘alternatives preys’ into ‘alternative prey’
Line 38: Calcium should not be written with a capital, just ‘calcium’

Introduction
Line 51: change ‘Sothern’ into ‘Southern’
Line 62: change ‘optimum’ into ‘optimal’
Line 62: remove ‘high’
Line 75: ‘depends on limited and overexploited fisheries’. The fisheries is finite (limited) but it is not reasonable to call all the fisheries overexploited. Moreover, a substantial part of the fish meal and fish oil is now coming from fish slaughter waste.
Line 94-96: Furthermore, ….in aquaculture. In this senctence ‘elements’ would be a better word than ‘metals’. Also, the suitability of the prey is not evaluated, it is their nutritional composition that is evaluated. Please change.

M&M
Line 123: I presume the samples were weighed before and after burning so include ‘gravimetrically’ before determined.

Statistical analysis:
Lines 190-191: Sentence can be taken away

Discussion
The discussion starts with the argument of overfishing, but that FAO report does not distinguish between fisheries for human consumption and for fish meal production. The argument of fisheries having (over)reached its limits and fish meal being a finite resource, meaning that if we want to increase the fish feed production we have to find alternatives makes a scientifically more correct argument.

Lastly, the authors suggest this as a possibility for extensive fish farming. It would be nice to see a few lines on the possible environmental impacts of this type of farming when it will find a market since it will put more pressure on the marshlands (or not?).

---

## Round 0.2 · accepted · Accept

The MS has been revised taking into account most of the comments of the referees.

There is still one point that I would like to suggest (which you can resolve in production). Regarding Nutrient requirements of fish, the proper / correct reference should be NRC 2011 and not the past one of 1993.